# Nanocrystalline ZnSnN_2_ Prepared by Reactive Sputtering, Its Schottky Diodes and Heterojunction Solar Cells

**DOI:** 10.3390/nano13010178

**Published:** 2022-12-30

**Authors:** Fan Ye, Rui-Tuo Hong, Yi-Bin Qiu, Yi-Zhu Xie, Dong-Ping Zhang, Ping Fan, Xing-Min Cai

**Affiliations:** Key Laboratory of Optoelectronic Devices and Systems of Ministry of Education and Guangdong Province, and Shenzhen Key Laboratory of Advanced Thin Films and Applications, College of Physics and Optoelectronic Engineering, Shenzhen University, Shenzhen 518060, China

**Keywords:** ZnSnN_2_, nanocrystalline, Schottky diode, heterojunction, solar cell

## Abstract

ZnSnN_2_ has potential applications in photocatalysis and photovoltaics. However, the difficulty in preparing nondegenerate ZnSnN_2_ hinders its device application. Here, the preparation of low-electron-density nanocrystalline ZnSnN_2_ and its device application are demonstrated. Nanocrystalline ZnSnN_2_ was prepared with reactive sputtering. Nanocrystalline ZnSnN_2_ with an electron density of approximately 10^17^ cm^−3^ can be obtained after annealing at 300 °C. Nanocrystalline ZnSnN_2_ is found to form Schottky contact with Ag. Both the current *I* vs. voltage *V* curves and the capacitance *C* vs. voltage *V* curves of these samples follow the related theories of crystalline semiconductors due to the limited long-range order provided by the crystallites with sizes of 2–10 nm. The *I*−*V* curves together with the nonlinear *C^−2^*−*V* curves imply that there are interface states at the Ag-nanocrystalline ZnSnN_2_ interface. The application of nanocrystalline ZnSnN_2_ to heterojunction solar cells is also demonstrated.

## 1. Introduction

The family of Zn-IV-N_2_ (IV = Si, Ge and Sn) is an analogue of the III nitrides [1]. As a member of this family, ZnSnN_2_ has many advantages, which include a direct band gap, earth-abundance of constituent elements, no toxicity, a high absorption coefficient, a low fabrication cost, etc. [1,2,3,4,5,6]. These make ZnSnN_2_ a highly competent candidate in photocatalytic, photovoltaic and light-emitting applications. Recent theoretical work shows that the photon-to-electron conversion efficiency of ZnSnN_2_/CuCrO_2_ heterojunction solar cells is approximately 22% [2].

ZnSnN_2_ has been fabricated by various methods such as sputtering [3,4,5,6,7,8,9,10,11,12,13,14], molecular beam epitaxy (MBE) [15], high pressure metathesis reaction [16] and vapor-liquid-solid growth [17]. During the preparation of ZnSnN_2_, Sn atoms can easily occupy the positions of Zn atoms, and this results in the formation of the intrinsic antisite defect Sn_Zn_ which is a divalent donor and which almost has the lowest formation energy in ZnSnN_2_ [18,19]. Therefore, the antisite defect Sn_Zn_ is considered to be the major donor in most cases, and the prepared ZnSnN_2_ usually has a very high density of approximately 10^19^ cm^−3^. Preparing ZnSnN_2_ with an electron density of approximately or below 10^17^ cm^−3^ is still challenging due to the facile formation of the donor defect Sn_Zn_.

Compared with its polycrystalline, crystalline or microcrystalline counterparts [3,4,5,6,7,8,9,10,11,12,13,14,15,16,17], nanocrystalline ZnSnN_2_ whose crystalline grains or crystallites are smaller than 10 nm has never been studied. In this paper, we approach the high electron density problem of ZnSnN_2_ with nanocrystallization. We show that nanocrystalline ZnSnN_2_ with crystallites of approximately 2 nm in the amorphous matrix can be deposited by sputtering an alloy target of Zn and Sn (the Zn/Sn atomic ratio to be 3:1) in an atmosphere of N_2_ and Ar. Annealing at 300 °C makes the reactively deposited ZnSnN_2_ turn *n*-type conductive with an electron density of approximately 10^17^ cm^−3^ and the annealed ZnSnN_2_ remains nanocrystalline with a grain size of 2–10 nm. The device applications, such as Ag-ZnSnN_2_ Schottky diodes and Cu_2_O-ZnSnN_2_ heterojunction solar cells, are studied.

## 2. Materials and Methods

### 2.1. Preparation and Characterization of ZnSnN_2_

Reactive radio frequency (RF) magnetron sputtering was used to deposit the samples. The substrates were K9 glasses. The substrates were ultrasonically cleaned in acetone, ethanol and deionized water, and the washing time in each liquid was 15 min. The target was the alloy of Zn (99.99%) and Sn (99.99%), with the Zn/Sn atomic ratio to be 3:1, and the gases were Ar (99.99%) and N_2_ (99.999%). The flow rate of Ar was 20 standard cubic centimeters per min (sccm), and that of N_2_ was 6 sccm. The base pressure of the sputtering chamber was 5.4 × 10^−4^ Pa. During sputtering, the chamber pressure was 5 Pa and the RF sputtering power was 35 W. The substrates rotated with the substrate holder at 0.6π rad/s. Before film deposition, the target was sputtered for 5 min to clean the surface. The time for film deposition was 60 min. Some deposited samples were annealed for 1 h in a vacuum chamber in the flow of Ar (20 sccm) and at a pressure of 5 Pa. The annealing temperatures were 300 °C, 350 °C, 400 °C and 450 °C.

The films without annealing or annealed at different temperatures were characterized with X-ray diffraction (XRD, D/max 2500 PC, 18 kW, Cu Kα radiation) and transmission electron microscopy (TEM, FEI Titan Cubed Themis G2 300). The samples for TEM observation were prepared by using the mechanical stripping method. The film thickness was measured with the surface profiler (Veeco Dektak 3ST), and the surface of the samples was characterized with scanning electron microscopy (SEM, Supra55 Sapphire). The samples were also characterized with atomic force microscopy (AFM, Oxford Instrument, MFP-3D Infinity, noncontact mode), room temperature Hall effect (HL 5500 PC, Van der Pauw electrode was used, and melted alloy of In and Sn was dropped at the four corners of a square sample to act as the electrodes) as well as X-ray photoelectron spectroscopy (XPS, Thermo Fisher, Thermo escalab 250Xi, Al Kα = 1486.6 eV; The C 1 s was calibrated to be 284.8 eV; the calculation of the atomic ratio is elaborated elsewhere [14]).The Seebeck coefficient *S_e_* of the samples was measured with Δ*V*/Δ*T* where Δ*V* is the voltage difference (Δ*V* is obtained with a voltmeter (Keithley 2400)) and Δ*T* is the temperature difference between the hot side and the cold side.

### 2.2. Preparation and Characterization of the Schottky Diodes and Heterojunctions

The steps to prepare the Schottky diodes with the structure of Ag\ITO\ZnSnN_2_\Ag (ITO refers to indium tin oxide, and its resistivity is approximately 6.22 × 10^−4^ Ω·cm) are as follows (Appendix A): ZnSnN_2_ was first deposited on ITO (whose substrate is glass) without annealing, and the thickness was 95 nm. During film deposition, some surface area of the ITO was covered with a mask in order to produce electrodes later. Secondly, a silver paste of 0.1 cm × 0.1 cm was dropped on ZnSnN_2_ and the surface of ITO to act as electrodes. To compare the effect of annealing, ZnSnN_2_ deposited on ITO under the same conditions was annealed for 1 h at 300 °C in the flow of Ar before making the electrodes. During annealing, the vacuum pressure was also 5 Pa, and then the second step of dropping silver paste was repeated to produce electrodes for another Schottky diode.

To prepare Cu_2_O-ZnSnN_2_ heterojunctions (Appendix A), Cu_2_O was first deposited on ITO with reactive DC sputtering (part of the ITO was also masked for making an electrical connection later). The sputtering voltage and current were 289 V and 35 mA. The target was copper (99.999%), and the gases were Ar (30 sccm) and O_2_ (1.8 sccm). The work pressure was 0.6 Pa, and the substrate temperature was 400 °C. The sputtering time was 10 h, and the thickness of Cu_2_O was 1672 nm (to obtain the electrical properties of Cu_2_O, Cu_2_O was also deposited on K9 glass, and Hall measurements showed that Cu_2_O is *p*-type conductive with a hole density of 1.02 × 10^15^ cm^−3^). After Cu_2_O cooled to room temperature naturally, ZnSnN_2_ was deposited on Cu_2_O and the parameters were the same as those mentioned previously except that the sputtering time was 3 h here (the thickness of ZnSnN_2_ was 236 nm here). Silver paste was dropped on the surface of ITO and ZnSnN_2_ to produce one solar cell with the structure of Ag\ITO\Cu_2_O\ZnSnN_2_\Ag. Additionally, with silver paste dropped on the surface of ITO to produce an electrode, Au was evaporated on ZnSnN_2_ to produce another solar cell with the structure of Ag\ITO\Cu_2_O\ZnSnN_2_\Au. The base pressure for evaporating Au was 5 × 10^−4^ Pa. Au wire (99.999%) was heated first with a current of 75 A for 10 min and then at a current of 90 A for 5 min. The solar cells had an area of 0.04 cm^2^.

The current *I*-voltage *V* curves of the devices were measured at room temperature with a source measurement unit instrument (Keithley Standard Series 2400). The capacitance *C*-voltage *V* curves of the devices were measured at room temperature with a semiconductor characterization system (Keithley 4200-SCS), and the frequency used was 10 kHz. For the Schottky diodes, the forward-bias is defined as the state at which the Ag electrode, in direct contact with the ZnSnN_2_ layer, is connected with the anode of the voltage. The solar cells are forward-biased when the electrostatic potential of Cu_2_O is higher than that of ZnSnN_2_. The current density-voltage V curves of the two Cu_2_O-ZnSnN_2_ heterojunction solar cells were measured at room temperature with a multi-meter (Keithley, 2400 Series) under AM1.5 light illumination (which is from an AAA solar simulator with the intensity calibrated to 100 mW/cm^2^ through a Si reference cell).

## 3. Results and Discussion

The XRD patterns of the samples deposited at 35 W and room temperature (the substrates were not intentionally heated) without annealing and those deposited at the same conditions but annealed at 300–450 °C are presented in Figure 1a. No diffraction peaks were observed in the XRD patterns of these samples. The samples deposited at 35 W and room temperature without annealing were amorphous, and they were still amorphous after annealing at 300–450 °C. In our equipment, when the sputtering power and other parameters were kept unchanged, the samples deposited at the substrate temperature of 100–300 °C became polycrystalline, and the preferred orientation changed when the substrate temperature was over 200 °C (Figure 1b, also Appendix A). However, when the sputtering power is 50 W, even the samples deposited at room temperature are polycrystalline (Figure 1c; also Appendix A). This implies that relatively higher sputtering power favors crystallization, and this agrees with F. Alnjiman et al., who showed that polycrystalline ZnSnN_2_ can be deposited without heating the substrate intentionally [7]. Though the samples deposited at the substrate temperature of 100–300 °C were polycrystalline, the samples deposited at room temperature and then annealed at 300 °C or above after deposition remained amorphous, and post-deposition annealing failed to make the amorphous samples turn to polycrystalline. This is different from amorphous ZnSnN_2_ prepared by direct current sputtering, which turns into polycrystalline after annealing [12]. Therefore, amorphous ZnSnN_2_ is relatively stable since it remains amorphous even if the post-deposition annealing temperature is higher than 300 °C. It is reported that reducing the Zn/Sn ratio to a certain degree leads to microcrystalline and amorphous ZnSnN_2_ [13], while in our work, reducing the substrate temperature to fabricate amorphous ZnSnN_2_ was used. Later, we will show that both the ZnSnN_2_ samples without annealing and the ZnSnN_2_ samples annealed at 300 °C are actually nanocrystalline with crystallites of 2–10 nm. The Raman scattering spectra of these samples show that the obtained ZnSnN_2_ is phonon-glass-like (Appendix A).

The transmittance (*T_r_*) and reflectance (*R*) spectra of the samples were measured with an UV/VIS spectrophotometer. The absorption coefficient *α* is equal to $t^{-1}\ln[(1-R)T_{r}{}^{-1}]$, where *t* is the thickness of the film (cm). The thickness of the films before and after annealing is approximately 95 nm. For semiconductors with a direct band gap, the relation between the absorption coefficient *α* and the optical band gap $E_{g}^{opt}$ is as follows:(1)$${\left(\alpha h\nu \right)}^{2}=A(h\nu -E_{g}^{opt}),$$
where *A* is a constant, *h* is the Plank constant and *ν* is the photon frequency. The optical bandgap $E_{g}^{opt}$ can be obtained by extrapolating the linear region to intercept the *hν* axis in the plot of ${\left(\alpha h\nu \right)}^{2}$ vs. *hν*. With the assumption that nanocrystalline ZnSnN_2_ has a direct band gap, the optical bandgap $E_{g}^{opt}$ of all the samples can be obtained, as shown in Figure 2. The optical bandgap of nanocrystalline ZnSnN_2_ is approximately 2.75 eV, which is larger than the experimental band gap of crystalline or polycrystalline ZnSnN_2_ (0.94–2.38 eV [15,20]), and this is in agreement with the fact that the band gap of amorphous Si is larger than its crystalline counterparts. The optical band gaps of the samples annealed at 300 °C, 350 °C, 400 °C and 450 °C were 2.40 eV, 2.35 eV, 2.11 eV and 2.15 eV, respectively. In amorphous and microcrystalline ZnSnN_2_ fabricated by reducing the Zn/Sn ratio, the optical bandgap is also approximately 2.53–2.59 eV [13], and our work agrees with this. It has been found that annealing reduces the band gap.

Hall effect measurements were conducted, and the results of the samples are listed in Table 1. The samples without annealing are almost insulating, and the resistivity is beyond the measurement scope of the Hall effect instrument. The samples without annealing are still *n*-type conductive since their Seebeck coefficients *S_e_* were measured to be negative [21], and no reversal in the conduction type is observed here. After being annealed at 300–450 °C, the samples turn conductive. The electron density *n* of the samples annealed at 300 °C is 5.54 × 10^17^ cm^−3^ and this is much lower than *N_c_*, the room temperature density of the states of the conduction band of ZnSnN_2_ (1.04 × 10^18^ cm^−3^) [4,14]. The gap between the conduction band minimum *E_c_* and the Fermi level *E_F_* was calculated to be 0.0165 eV with
(2)$$n=N_{c}\exp[-\left(E_{c}-E_{F}\right)/\left(kT\right)],$$
where *k* is the Boltzmann constant and *T* is the absolute temperature (300 K here). The electron density increases to ~10^18^ cm^−3^ when the annealing temperature is 350, 400 or 450 °C. The mobility is similar to that of the amorphous and microcrystalline samples deposited by reducing the Zn/Sn ratio [13].

To reveal the microscopic change of the structure, the samples without annealing and the samples annealed at 300 °C were studied with TEM (Figure 3). Crystallites with a size slightly larger than 2 nm can be observed, and no diffraction patterns are observed in the samples without annealing (Figure 3a). The fact that no diffraction patterns are observed possibly results from the fact that the density of the crystallites or the volume ratio of the crystalline component to the amorphous component is small (Figure 3a). After being annealed at 300 °C, crystallites with bigger sizes, larger densities and diffraction spots are observed in the high-resolution image and diffraction patterns, and the sizes of the crystallites are 5–10 nm (Figure 3b). Annealing is found to increase the size of crystallites. In amorphous Si with crystallites smaller than 20 Å, a reversal in the conduction type can be observed in the Hall effect measurement [22,23]. The fact that no reversal in the conduction type is observed in the Hall effect measurement of our samples very possibly results from the nanocrystalline grains, which provide limited long-range order [22].

The AFM images of the samples fabricated at 50 °C, 100 °C, 250 °C and 300 °C and at 35 W are presented in Figure 4. The root mean square (RMS) roughness of the samples deposited at 50 °C, 100 °C, 250 °C and 300 °C is 0.77 nm, 1.27 nm, 0.92 nm and 2.35 nm, respectively. The amorphous samples deposited at 50 °C are found to have the lowest RMS roughness. The morphology of polycrystalline samples is similar to those obtained by others [24].

The samples were measured with XPS. A survey in 0–1400 eV shows that the samples contain Zn, Sn and N, together with adventitious O. The high-resolution binding energy spectra of Zn, Sn, N and O of the samples without annealing and those annealed at 300 °C are presented in Figure 5a,b.

For the ZnSnN_2_ samples without annealing, Zn 2p3/2 and Zn2p1/2 are at 1021.38 and 1044.45 eV, while for the ZnSnN_2_ samples annealed at 300 °C, these peaks shift to 1021.57 and 1044.64 eV. Therefore, Zn is at +2 [24]. The peak at approximately 498 eV present in the Sn spectrum is due to the Zn L3M45M45 Auger peak [4,25]. Sn is at +4 since Sn 3d5/2 and 3d3/2 are at 486.11 and 494.52 before annealing, and these peaks shift to 486.18 and 494.58 after annealing [13,24]. In the high resolution XPS spectrum of N 1s, a peak at approximately 403 eV is observed and it is from absorbed γ-type nitrogen molecules labeled as γ-N_2_ (N ≡ N). The possible origin of γ-N_2_ is nitrogen molecules, which are not decomposed but absorbed directly during sputtering [26,27]. The peak below 400 eV can be deconvoluted into two peaks. In the samples without annealing, the two N 1 s peaks are at 396.31 and 397.80, and they shift to 396.90 and 398.35 eV after annealing. The lower energy peak with a much stronger intensity at 396.31 and 396.90 eV implies nitrogen is at −3, while the weaker higher energy peak at 397.80 and 398.35 eV is from organic nitrogen [7,13].

The O 1s peak can also be decomposed into two peaks. The peaks of the samples without annealing are at 529.90 and 531.18 eV, and after annealing, the peaks shift to 529.99 and 530.58 eV. The exact origin of the oxygen present in ZnSnN_2_ is still under debate. Some researchers think that the oxygen present in ZnSnN_2_ is completely from post-deposition adsorption in air [7]. In our work, we think that the oxygen has two possible origins: the low energy peak is possibly from residual oxygen in the sputtering chamber, which was incorporated during sputtering and which substitutes nitrogen in the lattice [13]; the high energy peak is due to post-deposition adsorption in air.

The atomic ratio of Zn to Zn + Sn is 0.80 before annealing and 0.78 after annealing. In the literature, ZnSnN_2_ samples with Zn/(Zn + Sn) = 0.72 and an electron density of 2.7 × 10^17^ cm^−3^ were reported [5]. The decrease is mainly due to the loss of Zn, since Zn has much higher vapour pressure than Sn at the annealing temperature of 300 °C. Most possibly, Sn atoms, which substitute the positions of Zn atoms, are the major donors in these nanocrystalline samples, and this is similar to its polycrystalline or crystalline counterpart [18,19].

Figure 6 shows the current density *J* vs. voltage *V* curves of the Schottky diodes with the structure of Ag\ITO\ZnSnN_2_\Ag. The diode is forward-biased when the silver metal directly on the ZnSnN_2_ layer is connected with the anode of the voltage. Rectification is observed, and obviously the rectification is from the Schottky contact formed between ZnSnN_2_ and Ag since the structure of Ag\ITO\ZnSnN_2_\ITO\Ag shows linear *JV* curves (Appendix A). The transport properties of our nanocrystalline ZnSnN_2_ very possibly follow those of crystalline semiconductors since the transport behaviour of amorphous and microcrystalline Si with crystallites of or over 20 Å follows that of crystalline Si [22,23], and the crystallite of our nanocrystalline ZnSnN_2_ meets this size requirement. In the following, theories or models based on crystalline semiconductors will be used. Under the thermal emission-diffusion model, the current through a Schottky barrier diode at a voltage of *V* [28,29] is as follows:(3)J=Jsexp[−ξq(V−IRs)/(kT)]{exp[q(V−IRs)/(kT)]−1},
where Js=Is/S=A**T2exp[−qϕb0/(kT)], *J_s_* is the current density, *I_s_* is the saturation current at zero bias, *S* is the diode area (0.10 × 0.10 cm^2^ here), *A*^∗∗^ is the effective Richardson constant (*A*^∗∗^ = 14.40 A·cm^−2^·K^−2^ for ZnSnN_2_ since the Richardson constant *A** for a free electron is 120 A·cm^−2^·K^−2^, and the effective electron mass of ZnSnN_2_ is 0.12 times that of a free electron [14]), *q* is the elementary charge, *ϕ_b0_* is the barrier height at zero bias (the energy needed by an electron at the Fermi level in the metal to enter the conduction band of the semiconductor at zero bias), *R_s_* is the series resistance due to the neutral region of the semiconductors, *ξ* is a factor without unit, *ξ* is equal to 1 − *η*^−1^, where *η* is the ideality factor, and other symbols have the same meaning as in previous formulas. From the forward-biased experimental data, the three unknown parameters *ϕ_b0_*, ξ and *R_S_* can be fitted with the least square method, and the results, together with the calculated *η,* are presented in Figure 6a.

At lower forward biased voltages, the fitted current is slightly larger than the experimental data, while good fitting is observed when the forward bias is over ~0.3 V in both samples. Since the mobility of both samples is low, diffusion is most likely the major transport mechanism at lower forward biased voltages, and thermal emission becomes obvious with an increase in the forward biased voltage. The fitted series resistance *R_s_* of ZnSnN_2_ without annealing is 277.8 Ω, which is larger than that of ZnSnN_2_ annealed at 300 °C (9.2 Ω) and this agrees with the fact that the former has poorer conductivity than the latter.

The theoretical Schottky barrier height *ϕ_b0_* at zero bias for ideal or intimate Ag-ZnSnN_2_ is calculated to be 0.36 eV (the ideal band diagram is shown in Appendix A) since theoretical *ϕ_b0_* is equal to *W_m_ − χ,* where *W_m_* is the work function of Ag (*W_m_* = 4.26 eV for Ag) and *χ* is the electron affinity of ZnSnN_2_ (*χ* = 3.9 eV for ZnSnN_2_ [30,31]). The fitted *ϕ_b0_* is larger than the theoretical value, and the Schottky barrier height is enhanced. Schottky barrier height enhancement can result from surface Fermi-level pinning [32], a surface inversion layer (which possibly results from the diffusion of Ag into ZnSnN_2_, and Ag atoms substituting Zn or Sn are acceptors) [33,34] or interface oxide states [32]. Surface Fermi-level pinning is less likely here since the forbidden band gap *E_g_* calculated from the fitted *ϕ_b0_* will be approximately 1.0 eV (under surface Fermi-level pinning, *ϕ_b0_* is equal to *E_g_* − *ϕ*_0_, where *ϕ*_0_ is the surface neutral energy level measured from the edge of the valence band and is usually approximately *E_g_*/3 [32]), which is less than half of those obtained from the Tauc plot. The fact that the calculated ideality factor *η* is over two also suggests the existence of interface oxide states since the ideality factor of a Schottky diode whose barrier height is enhanced by a surface inversion layer is usually below two [33].

A. M. Cowley and S. M. Sze [32] proposed that when there is interfacial oxide (the band diagram of the Schottky contact with interface states is shown in Appendix A), the Schottky barrier height is expressed as:(4)$$\phi_{b0}=\gamma (W_{m}-\chi )+(1-\gamma )(E_{g}-\phi_{0})/q-\Delta \varphi_{n}$$
and the interface states $D_{s}=(1-\gamma )\varepsilon_{i}/(\gamma q^{2}\delta )$, where *γ* is a weighting factor (without unit), $\Delta \varphi_{n}$ is the image force barrier lowering, *ε_i_* is the dielectric constant of the interfacial layer, *δ* is its thickness and other symbols have the same meaning as previously defined. If we suppose $\Delta \varphi_{n}=0$, $\phi_{0}=E_{g}/3$, where *E_g_* is chosen to be 2.5 eV and *ϕ_b0_* = the fitted value, *γ* will be 0.71 for the samples without annealing and 0.83 for the samples annealed at 300 °C. With the assumption that *ε_i_* = *ε_0_* (free space dielectric constant) and *δ* = 5 Å [32], the interface states *D_s_* will be 4.42 × 10^12^ states·eV^−1^·cm^−2^ for the samples without annealing and 2.19 × 10^12^ states·eV^−1^·cm^−2^ for the samples annealed. The samples without annealing have larger interface states than the samples annealed. The interface states of our samples are in the same magnitude with that of the Al/*n*-Si system (where Si is single-crystalline) [35], but roughly one magnitude smaller than that (3 × 10^13^ states·eV^−1^·cm^−2^) of metals and amorphous Si systems which have the same oxide thickness of 5 Å and whose Fermi-level were pinned [36].

Figure 6b,c shows the logarithmic J-V curves of the samples without annealing or annealed at 300 °C. For both samples, both the forward- and reverse-biased data obey the power law ($J\propto V^{m}$), but m varies in different voltage ranges. For the samples without annealing, below 0.18 V, m is less than one, and this suggests that the current is controlled by the diode. Between 0.18 V and 0.68 V, m is approximately 19.2. After 0.68 V, m is approximately 230. This implies that after 0.18 V, the current is bulk-controlled, and there is an exponential distribution of trap-levels within the forbidden gap of ZnSnN_2_. The reverse-biased data of the samples without annealing is diode-controlled since m is below one in the whole measurement range. For the samples annealed at 300 °C, m is well over two in both the forward and backward bias, and this suggests that the current is bulk-controlled, and annealing reduces the resistivity of ZnSnN_2_.

Figure 7a,b show the curves of the inverse of the square of the capacitance *C* (per unit area) vs. voltage *V* for the samples without annealing or annealed at 300 °C. For both samples, nonlinear *C^−2^*-*V* curves are observed. The peak in Figure 7b is possibly due to the series resistance [37,38]. Considering the nonlinearity of the *C^−2^*-*V* curves and the fact that the ideality factor *η* obtained from the *IV* curves is over two, we think the possibility that the barrier height enhancement results from a surface inversion layer can be excluded [33]. The nonlinearity of the *C^−2^*-*V* curves suggests the existence of excess capacitance. When there is an interfacial layer in the Schottky contact, the capacitance *C* of a Schottky diode [39] is equal to ${\left[C_{i}^{-1}+{\left(C_{D}+C_{ex}^{}\right)}^{-1}\right]}^{-1}$, where *C_i_* is the capacitance due to the interfacial layer, *C_D_* is the depletion capacitance of the junction and *C_ex_* is the excess capacitance due to surface states or deep traps. For thin interfacial layers, *C_i_* is very large and can be ignored. Therefore, *C* is equal to $C_{D}+C_{ex}^{}$. Though *C*^−2^ vs. *V* curves are not linear, (*C − C_ex_*)^−2^ vs. *V* or $C_{D}^{-2}$ vs. *V* will be linear.

For the Schottky barrier where there is an oxide between the metal and the semiconductor, J. Szatkowski et al. [40] further proposed that
(5)$$C=a(\lambda ){\left(V_{0}-V\right)}^{-0.5}+b(\lambda ),$$
where *a*(*λ*) is $(1-\lambda +\lambda \gamma ){\left(q\varepsilon_{0}\varepsilon_{r}N_{D}\gamma /2\right)}^{-0.5}$, *N_D_* is the doping concentration, *b*(*λ*) is $\lambda \gamma q^{2}D_{s}$, *V*_0_ and *λ* are constants, *ε_r_* is the relative dielectric constant of ZnSnN_2_ (*ε_r_* = 11 [20]) and other symbols have the same meaning as in previous formulas. From the data about the capacitance *C* and reverse-biased voltage *V*, *a*(*λ*), *b*(*λ*) and *V*_0_ can be fitted with the least square method. Both the experimental and fitted capacitance *C* vs. ${\left(V_{0}-V\right)}^{-0.5}$ curves (where only the reverse-biased data are used) of the samples without annealing or with annealing are shown in Figure 7c,d. The fitted *a*(*λ*), *b*(*λ*) and *V*_0_ are also presented in Figure 7c,d. As can been seen from Figure 7c,d, the relation between the capacitance *C* and ${\left(V_{0}-V\right)}^{-0.5}$ is indeed linear.

If *N_D_* is known, *λ* can be calculated from *a*(*λ*) and then *D_s_* from *b*(*λ*), since *γ* has been obtained from the *JV* curves, and *a*(*λ*) and *b*(*λ*) have been obtained from the *CV* curves. For the samples annealed at 300 °C, *λ* and *D_s_* are calculated to be 5.86 and 1.25 × 10^11^ states·eV^−1^·cm^−2^ if we suppose that *N_D_* = *n* = 5.34 × 10^17^ cm^−3^. The samples without annealing are almost insulating, and we can suppose that *N_D_* = 10^15^ cm^−3^ [36]. *λ* and *D_s_* are then calculated to be 2.49 and 3.40 × 10^11^ states·eV^−1^·cm^−2^. The interface states extracted from the capacitance-voltage of the samples without annealing are larger than those of the samples annealed at 300 °C, and this is in agreement with that obtained from *I*-*V* curves.

In addition, the interface states extracted from the *C*-*V* curves are roughly one-magnitude smaller than those from the *J*-*V* curves. The *J-V* and *C-V* curves of the Schottky diodes formed between nanocrystalline ZnSnN_2_ and Ag are found to follow the theories based on crystalline semiconductors, in agreement with amorphous or microcrystalline Si with crystallites between 20–130 Å [22,23].

Figure 8a shows the current density *J* vs. the voltage *V* curves of the Ag\ITO\Cu_2_O\ZnSnN_2_\Ag and Ag\ITO\Cu_2_O\ZnSnN_2_\Au heterojunction solar cells under the illumination of AM1.5. The only difference between the two solar cells is that the electrode in connection with ZnSnN_2_ is Ag for one solar cell and Au for the other. The open-circuit voltage *V_oc_* (V), short-circuit current density *J_sc_* (mA/cm^2^), fill factor FF and power conversion efficiency PCE (%) are listed in the inset. The statistical distribution of these solar cells is presented in Appendix A (the statistics are based on nine samples for the Ag\ITO\Cu_2_O\ZnSnN_2_\Au solar cell, while the statistics are based on eight samples for the Ag\ITO\Cu_2_O\ZnSnN_2_\Ag solar cell). The average *V_oc_*, *J_sc_*, FF and PCE of the solar cell with Au as the electrode are 0.22 V, 1.75 mA/cm^2^, 42% and 0.15%, while those of the solar cell with Ag as the electrode are 0.19 V, 0.34 mA/cm^2^, 22% and 0.02%, respectively. When the electrode in connection with ZnSnN_2_ changes from Ag to Au, the four parameters increase greatly, implying better device performance. The measured series resistance of the solar cell with Au as the electrode is 0.83 kΩ, and that with Ag as the electrode is 4.19 kΩ. The improvement in performance mainly results from the change in the electrode. Previous work shows that Au can react with Sn to form AuSn even at room temperature, and the contact between AuSn and *n*-type GaAs is ohmic [41]. Though the contact between Au and ZnSnN_2_ is theoretically non-ohmic (the work function of Au is 5.1 eV and the Fermi level of Au is much lower than that of ZnSnN_2_), the Au electrode was thermally evaporated, and it is possible that there are interfacial alloys formed such as AuSn that reduce the Au-ZnSnN_2_ contact resistance greatly.

Figure 8b shows the capacitance C-Voltage V curve of the Ag\ITO\Cu_2_O\ZnSnN_2_\Au heterojunction solar cell (measured under darkness), with the inset showing the *C^−2^-V* curve under reverse bias. It can be found that *C^−2^-V* in the reverse bias is linear, and the intercept between the *C^−2^-V* line and the voltage axis is approximately 0.24 V. This suggests that Cu_2_O-ZnSnN_2_ heterojunction is abrupt, with a built-in potential of 0.24 V. The built-in potential of 0.24 V is very near to the average open-circuit voltage (0.22 V) of the solar cell with Au as the electrode. Figure 8c shows the energy band diagram of the Cu_2_O-ZnSnN_2_ heterojunction at zero bias. The diagram shows a staggered, type II heterojunction. The built-in potential *V_D_*, conduction band and valence band discontinuities, Δ*E_c_* and Δ*E_v_*, are estimated to be 0.76 V, 0.80 eV and 1.2 eV, respectively (Appendix A). The obtained *V_oc_* needs great improvement as compared with *V_D_*. The presence of the conduction band and valence band discontinuities, Δ*E_c_* and Δ*E_v_*, accounts partially for the low PCE.

In addition, after being stored in air without encapsulation for 11 months, the JV curves of the Ag\ITO\Cu_2_O\ZnSnN_2_\Au solar cells were measured under the AM1.5 illumination and the average V_oc_, J_sc_, FF and PCE (%) were 0.06 V, 0.56 mA/cm^2^, 27% and 0.01%, respectively (the statistical data obtained after the storage for 11 months are also presented in Appendix A). The performance degradation might be due to the storage in air without encapsulation. Though all performance factors decrease as compared with those obtained 11 months ago, the photovoltaic effect still exists.

Figure 9a shows the dark JV curve on a log-log scale for the Ag\ITO Cu_2_O\ZnSnN_2_\Au heterojunction. The forward-biased JV curve obeys the power law (J∝Vm) with different exponent values m at different voltage ranges (the junction is forward-biased when the electrostatic potential of Cu_2_O is higher than that of ZnSnN_2_). In 0−0.19 V, m is smaller than unity, implying that the current is controlled by the heterojunction diode. This also suggests that the resistance of the heterojunction is not large, since large resistance usually results in ohmic behaviour under relatively low voltage. In 0.19–0.98 V, m is approximately two while over 0.98 V m is over three. This implies that the regime in 0.19–0.98 V is the trap-filled limited (TFL) region where only partial traps are filled and the current is due to a space charge limited current (SCLC) [42]. In the region over 0.98 V, all the traps are filled, strong injection happens and the current increases sharply with an increase in the voltage. If we suppose that V_tr_ (0.19 V in Figure 9a) is the turn-on voltage at which the space charge limited conduction takes place in ZnSnN_2_ and V_TFL_ (0.98 V in Figure 9a) is the voltage required to fill the traps of ZnSnN_2_, the trap density N_t_ and the trap energy level *E_t_* are calculated to be 2.14 × 10^16^ cm^−3^ and 0.1 eV, since V_tr_ equals 16qnt2Nt/{9εrε0Ncexp[(Et−Ec)/(kT)]} and V_TFL_ equals qNtt2/(2εrε0), where *n* is the free carrier density (*n* = *N_D_* = 10^15^ cm^−3^), *t* is the film thickness (236 nm) and the other symbols have the same meaning as previously defined. 

The heterojunction parameters, including the diode current *J*_0_, the ideality factor *η*, the series resistance *R_s_* and the shunt conductance *G* can be extracted based on a single diode model [43,44]:(6)$$J=J_{0}\exp[\frac{q}{\eta kT}(V-R_{s}^{}J)]+GV-J_{L},$$
where *J_L_* is the light-induced current (*J_L_* = *J_SC_* = 1.74 mA/cm^2^ here), while the other parameters have the same meaning as previously defined. Figure 9b shows the dJ/dV vs. V curve, and the shunt conductance G is found to be 0.43 mS/cm^2^ from the nearly flat area in the reverse bias. The dV/dJ vs. [*J* + *J_SC_*]^−1^ curve is presented in Figure 9c, and the series resistance *R_s_* is calculated to be 1.26 Ω·cm^2^. From the *J* + *J_SC_* − *GV* vs. *V* − *R_s_J* curve in Figure 9d, the diode current *J*_0_ and the ideality factor *η* are calculated to be 7.76 × 10^−2^ mA/cm^2^ and 2.90, respectively. The ideality factor *η* larger than two implies that the current transport is recombination-limited [45]. To improve the conversion efficiency, the series resistance needs to be reduced and the shunt resistance needs to be improved. The possible methods include better ohmic contact, optimum film thickness, etc. The band gap needs to be reduced to improve the short-circuit current density *J_sc_*. The theoretical open-circuit voltage *V_oc_* of the solar cell is estimated to be 0.24 V since *V_oc_* approximately equal $\left(\eta kT/q)\ln[(J_{sc}/J_{0})+1\right]$ if the series resistance and shunt resistance are negligible and the photogenerated current equals the short-circuit current density *J_sc_*. The diode current *J*_0_ needs to be reduced to further improve the open-circuit voltage *V_oc_*. 

The major structure and power conversion efficiency of several heterojunction solar cells, such as polycrystalline ZnSnN_2_, Si, GaAs, etc [12,30,46,47,48,49], are listed in Table 2. The PCE (%) of Cu_2_O-ZnSnN_2_ needs great improvement compared with these heterojunction solar cells. The low PCE of Cu_2_O-ZnSnN_2_ possibly mainly results from the band gap of nanocrystalline ZnSnN_2_ and the energy band diagram of the heterojunction. The band gap of nanocrystalline ZnSnN_2_ is approximately 2.5 eV and this is much wider than the optimum band gap to yield the highest PCE (%). A recent report shows that polycrystalline ZnSnN_2_ with a band gap of 1.43 eV can be prepared by sputtering under bias [50], and this provides a clue about reducing the band gap of nanocrystalline ZnSnN_2_. One possible method to reduce the conduction and valence band discontinuities (Figure 8c) is to deposit a buffer between Cu_2_O and ZnSnN_2_. Replacing Cu_2_O with other *p*-type semiconductors such as CuCrO_2_ is another method since theoretical work shows that the PCE (%) of CuCrO_2_-ZnSnN_2_ is approximately 22% [2]. ZnSnN_2_ belongs to the few solar cell absorption materials that can meet tetra Watt power needs at very low cost, and this necessitates further work in improving its material properties and device performance.

## 4. Conclusions

The reactive preparation and properties of nanocrystalline ZnSnN_2_, together with its Schottky diodes and heterojunction solar cells, were investigated. ZnSnN_2_ reactively deposited at 35 W and room temperature is nanocrystalline with crystalline grains of 2 nm. Annealing increases the grain size and the volume ratio of the crystalline component to the amorphous component. Annealing reduces the optical band gap and improves the electrical conductivity of ZnSnN_2_. Nanocrystalline ZnSnN_2_ annealed at 300 °C in Ar has an electron concentration of 5.54 × 10^17^ cm^−3^ and mobility of 0.998 cm^2^·V^−1^·s^−1^, which is much better than that of polycrystalline or crystalline ZnSnN_2_, whose electron density in most cases is usually approximately 10^19^ cm^−3^. The thermal emission-diffusion model was used to analyze the *I*-*V* curves of the Ag-nanocrystalline ZnSnN_2_ Schottky diode. At lower forward biased voltages, diffusion is the major transport mechanism, and thermal emission becomes obvious at higher forward biased voltages. Both the *I*-*V* and *C*-*V* curves suggest the existence of interface states, which are calculated to be approximately 10^12^ states·eV^−1^·cm^−2^ from the *IV* curves and approximately 10^11^ states·eV^−1^·cm^−2^ from the *CV* curves. Nanocrystalline ZnSnN_2_-based heterojunctions were successfully fabricated. The heterojunction interface is abrupt. The performance of the Ag\ITO\Cu_2_O\ZnSnN_2_\Au heterojunction solar cell is much better than that of the Ag\ITO\Cu_2_O\ZnSnN_2_\Ag solar cell. The current transport of the Ag\ITO\Cu_2_O\ZnSnN_2_\Au solar cell is recombination-limited. The fact that nanocrystallization can greatly reduce the electron density of ZnSnN_2_, the *IV* and *CV* curves of the Schottky contact formed between nanocrystalline ZnSnN_2_ and Ag obey the theories based on crystalline semiconductors and nanocrystalline ZnSnN_2_-based heterojunction solar cells are successfully prepared suggest that nanocrystallization can pave a new way for the device application of ZnSnN_2_.

## Figures and Tables

**Figure 1 nanomaterials-13-00178-f001:**
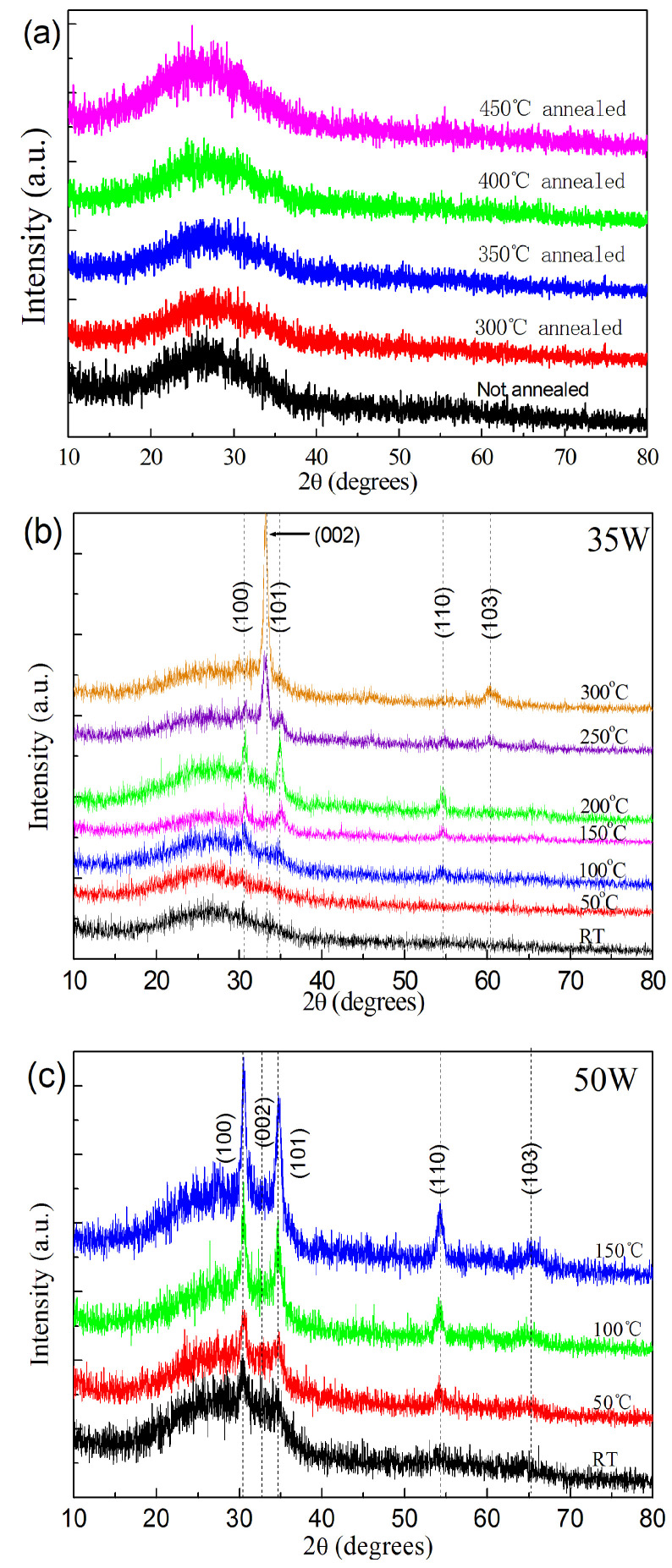
(**a**) XRD patterns of the samples without annealing or annealed at 300−450 °C; (**b**) XRD patterns of the samples deposited at 35 W and under different substrate temperatures; (**c**) XRD patterns of the samples deposited at 50 W and under different substrate temperatures.

**Figure 2 nanomaterials-13-00178-f002:**
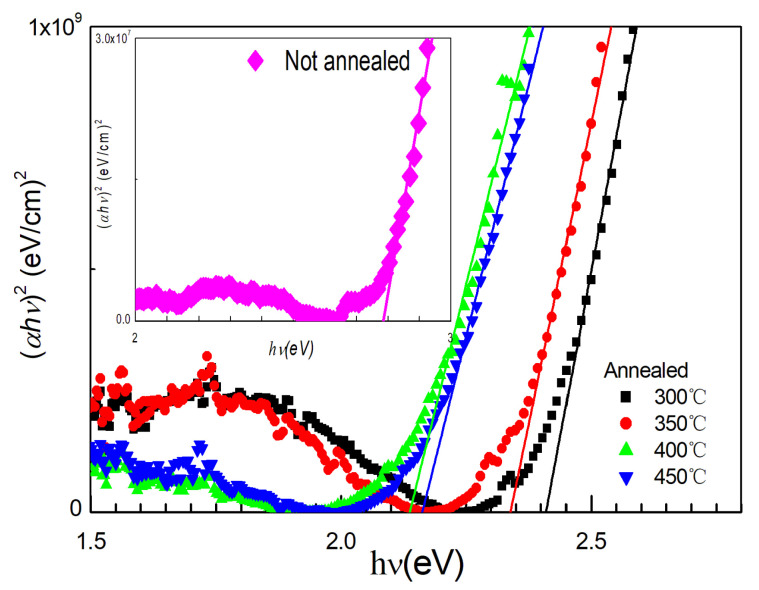
Tauc plots of ZnSnN_2_ without annealing or annealed at 300−450 °C.

**Figure 3 nanomaterials-13-00178-f003:**
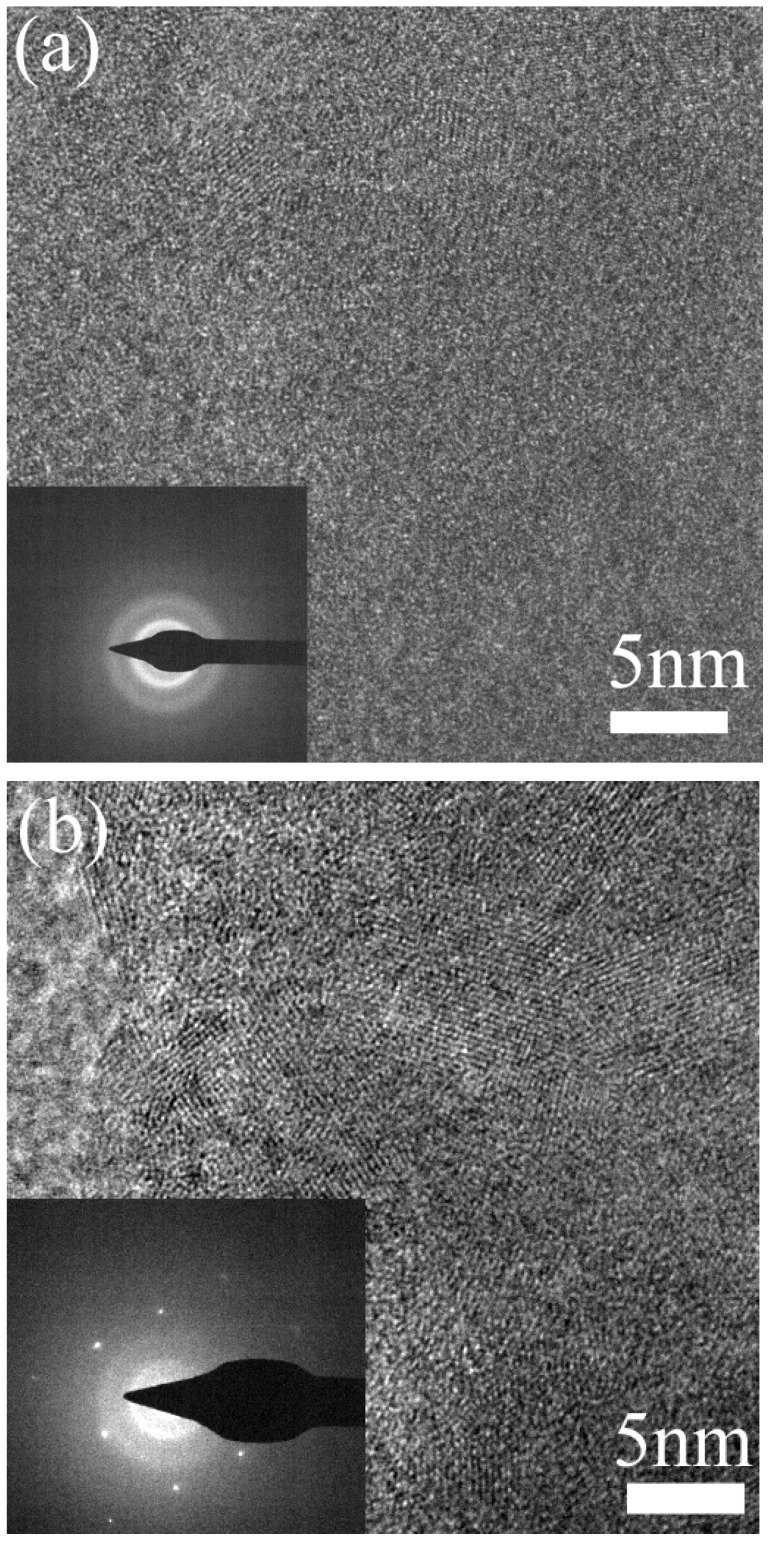
TEM images and the diffraction patterns of ZnSnN_2_. (**a**) Without annealing; (**b**) annealed at 300 °C.

**Figure 4 nanomaterials-13-00178-f004:**
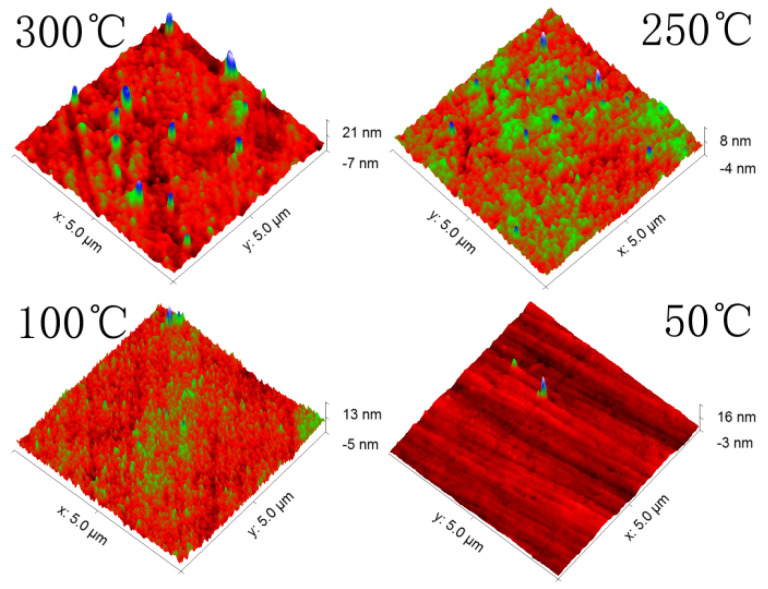
AFM images of the ZnSnN_2_ samples deposited at 35 W under different substrate temperatures.

**Figure 5 nanomaterials-13-00178-f005:**
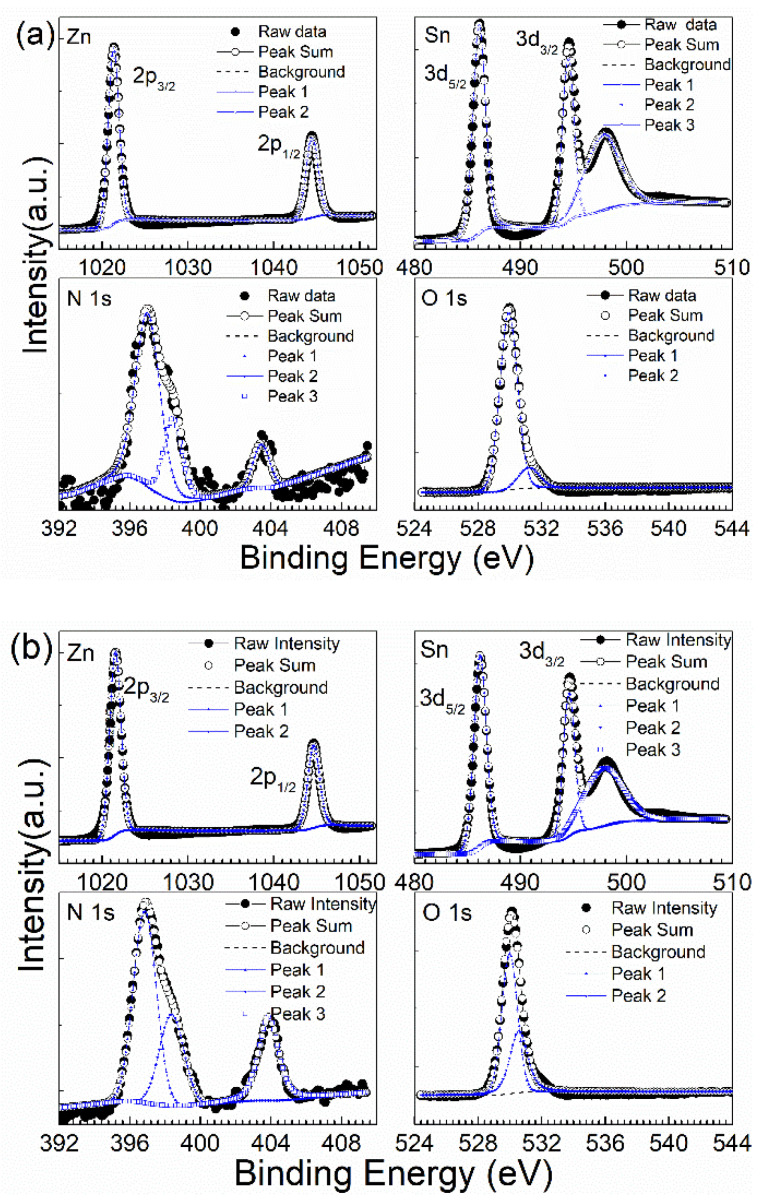
High resolution binding energy peak of Zn, Sn, N and O in ZnSnN_2_. (**a**) Without annealing; (**b**) annealed at 300 °C.

**Figure 6 nanomaterials-13-00178-f006:**
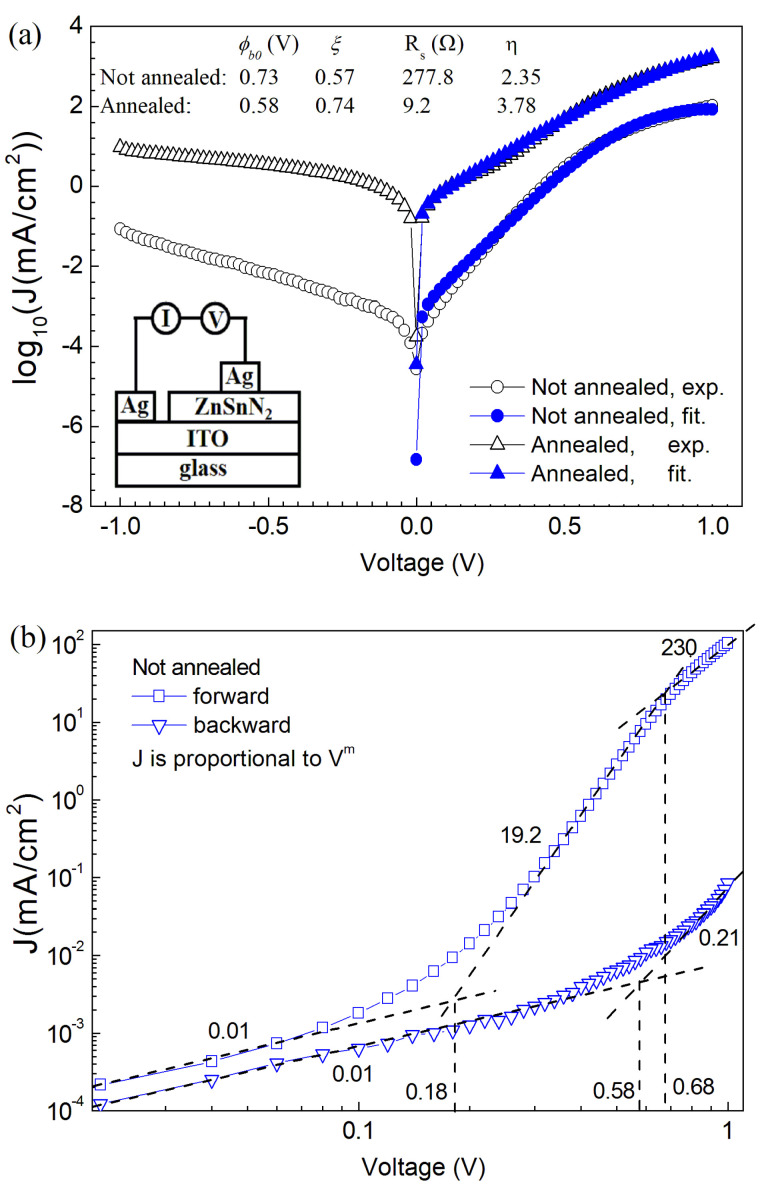
(**a**)The *J*−*V* curves of Ag\ITO\ZnSnN_2_\Ag Schottky contact (The Schottky contact is forward−biased when the Ag electrode in direct contact with ZnSnN_2_ is connected with the anode of the voltage. The inset shows the structure of the Schottky diodes). (**b**) log J−log V curves of the samples without annealing; (**c**) log J−log V curves of the samples annealed at 300 °C.

**Figure 7 nanomaterials-13-00178-f007:**
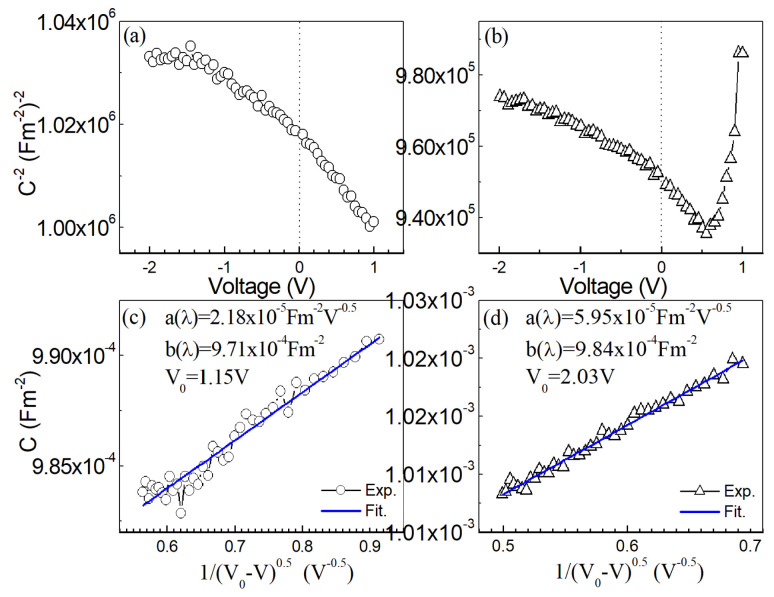
The *C*^−2^−*V* curves of the samples without annealing (**a**) or annealed at 300 °C (**b**); *C* − (*V*_0_ − *V*)^−0.5^ curves of the samples without annealing (**c**) or annealed at 300 °C (**d**).

**Figure 8 nanomaterials-13-00178-f008:**
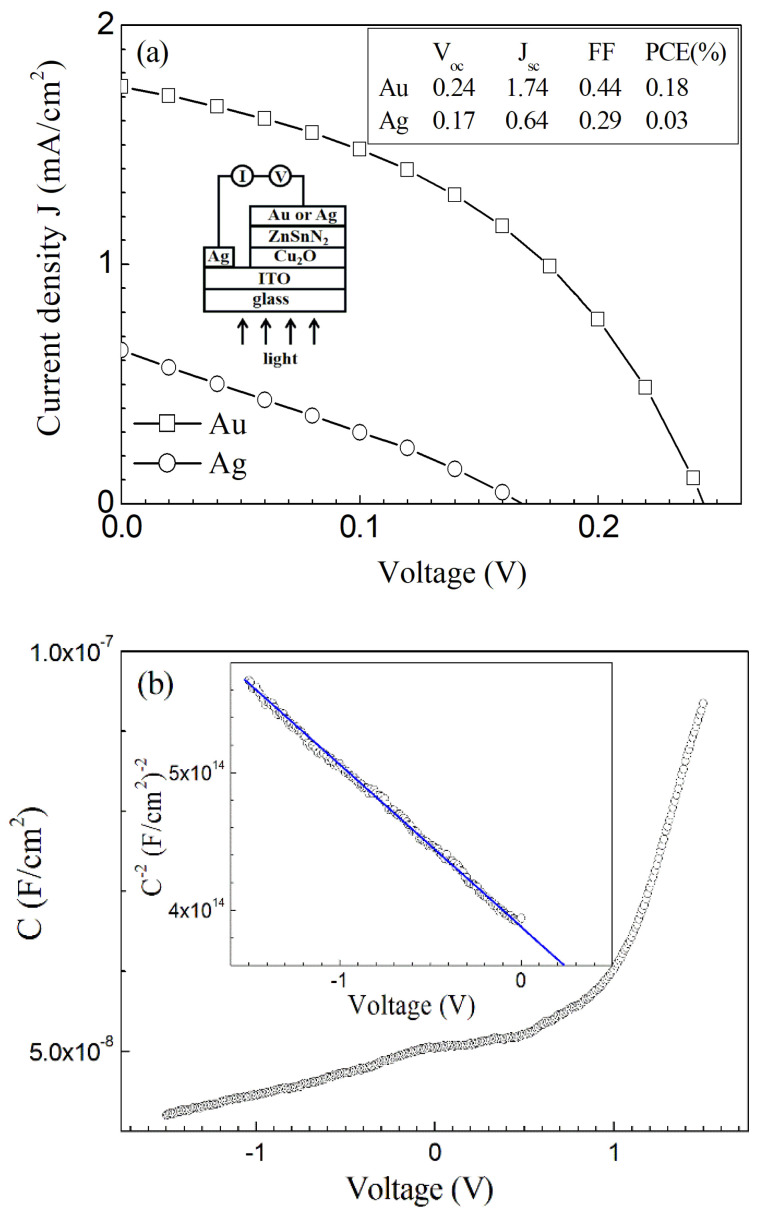
(**a**) The current density J vs. the voltage V curves of the Ag\ITO\Cu_2_O\ZnSnN_2_\Ag and Ag\ITO\Cu_2_O\ZnSnN_2_\Au solar cells (the inset shows the structure of these heterojunction solar cells); (**b**) the CV curve of the Ag\ITO\Cu_2_O\ZnSnN_2_\Au solar cell; (**c**) the energy band diagram of Cu_2_O−ZnSnN_2_ heterojunction at zero bias.

**Figure 9 nanomaterials-13-00178-f009:**
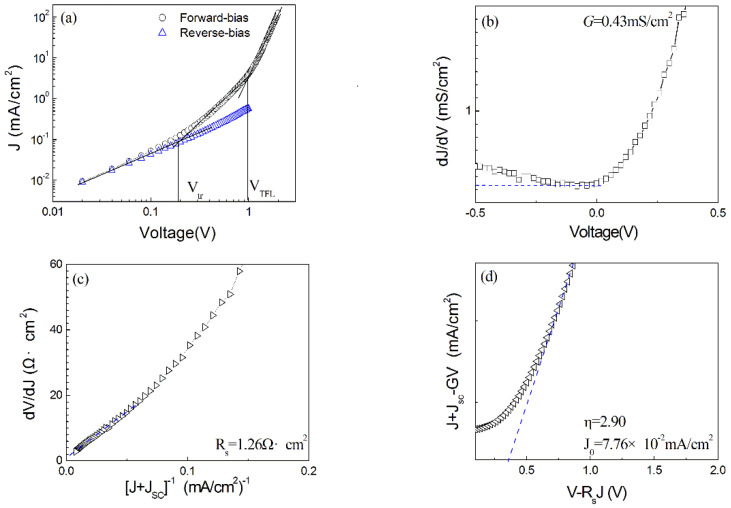
(**a**) The log−log dark JV curve; (**b**) dJ/dV vs. V curve; (**c**) dV/dJ vs. [*J* + *J_SC_*]^−1^ curve; (**d**) *J* + *J_SC_* − GV vs. V−RJ curve.

**Table 1 nanomaterials-13-00178-t001:** The mobility *µ* (cm^2^V^−1^s^−1^) and carrier concentration *n* (cm^−3^) of the samples annealed at 300–450 (°C) (The resistivity of the samples without annealing is beyond the scope of the instrument).

°C	*μ* (cm^2^V^−1^s^−1^)	*n* (cm^−3^)
300	0.988	5.34 × 10^17^
350	5.54	8.37 × 10^18^
400	2.96	6.48 × 10^18^
450	1.22	8.11 × 10^18^

**Table 2 nanomaterials-13-00178-t002:** The major structure and power conversion efficiency (PCE) of several heterojunction solar cells (pc refers to polycrystalline).

Layer/Stack	PCE (%)
pc-ZnSnN_2_/SnO and pc-ZnSnN_2_/Al_2_O_3_/SnO	0.37 [12] and 1.54 [30]
α-C/Si	1.5 [46]
ZnO/ZnGeO/Cu_2_O:Na	8.1 [47]
MXene/GaAs	9.69 [48]
α-Si:H/c-Si	25.6 [49]

## Data Availability

The data presented in this study are available upon request from the corresponding author. The data are not publicly available due to privacy.

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
