# Peer review of "Nanocrystalline ZnSnN2 Prepared by Reactive Sputtering, Its Schottky Diodes and Heterojunction Solar Cells"

_nanomaterials, 2022, doi:10.3390/nano13010178_

Round 1

Reviewer 1 Report

The zinc tin nitride films are considered potentially interesting for photovoltaic or photonic applications such as solar cells or emitters. Despite some ten years of research and relevant information obtained about the elaboration and the characterization of ZTN thin films, further research into the electrical and optical properties is much needed. The paper under review brings additional findings on the preparation of ZTN nanostructured films, their structure, optical and electrical properties. I would like to see this article published but after some minor modifications as follow:

1     1)  The preposition is required in the sentence “The samples were also characterized atomic force microscopy…” (page 2, lines 6-7 in paragraph 2);

2     2) The authors transferred the X-ray data on the samples deposited at different substrate temperatures to Supplementary Materials (Fig. S2). In my opinion, they have to present in the main text, and authors should explain why XRD pattern changes at a substrate temperature above 200 °C: instead of some peaks, others appear. And why ZTN turns out hexagonal (wurtzite?) instead of the usual orthorhombic.

3     3) When discussing the wider optical band gap of nanocrystalline ZTN compared to polycrystalline and crystalline ZTN (page 4), the possible quantum size effect is not taken into account. Why?

4     4) There is no Figure 5(a) to which authors refer in the text (page 9, top) discussing the experimental data on I-V curves of the Schottky diodes.

Authors should check English grammar, e.g. subject-verb matching in a sentence in some cases.

Author Response

Response to Reviewer 1

Comments and Suggestions for Authors

The zinc tin nitride films are considered potentially interesting for photovoltaic or photonic applications such as solar cells or emitters. Despite some ten years of research and relevant information obtained about the elaboration and the characterization of ZTN thin films, further research into the electrical and optical properties is much needed. The paper under review brings additional findings on the preparation of ZTN nanostructured films, their structure, optical and electrical properties. I would like to see this article published but after some minor modifications as follow:

1     1)  The preposition is required in the sentence “The samples were also characterized atomic force microscopy…” (page 2, lines 6-7 in paragraph 2);

Reply: "with" was inserted right before the word atomic.

2     2) The authors transferred the X-ray data on the samples deposited at different substrate temperatures to Supplementary Materials (Fig. S2). In my opinion, they have to present in the main text, and authors should explain why XRD pattern changes at a substrate temperature above 200 °C: instead of some peaks, others appear. And why ZTN turns out hexagonal (wurtzite?) instead of the usual orthorhombic.

Reply:

   Fig.S2 and Fig.S3 are transferred to the main text as Fig. 1(b) and Fig.1(c). Correspondingly, the original Fig.1 is now renamed as Fig.1 (a).

  Different substrate temperature results in different orientation. " and the preferred orientation changes when the substrate temperature is over 200 0C " was inserted to make an explanation.

  Theoretical work shows that ZnSnN2 is orthorhombic when Zn and Sn occupy the cation sublattice alternatively and hexagonal when the cation sublattice is disordered [Adv. Energy Mater. 5(2015)1501462; Phys. Rev. Mater.1(2017)034401; Phys. Status Solidi B254(2017)1600718]. Many diffraction peaks overlap for the orthorhombic and hexagonal structure. The major difference between orthorhombic and hexagonal structure is that there are diffraction peaks around 220 for the former[Adv. Energy Mater. 5(2015)1501462; Phys. Rev. Mater.1(2017)034401; Phys. Status Solidi B254(2017)1600718]. Though the diffraction peaks around 220 have never been experimentally observed, ZnSnN2 is indexed as either orthorhombic [Adv. Mater. 25(2013)2562] or hexagonal [Adv. Optical Mater.9(2021)2100015]. In our work, Raman scattering also demonstrates there is disorder in the cation sublattice. Therefore, our samples were indexed as hexagonal.

3     3) When discussing the wider optical band gap of nanocrystalline ZTN compared to polycrystalline and crystalline ZTN (page 4), the possible quantum size effect is not taken into account. Why?

 Reply: If we suppose that higher annealing temperatures results in crystallites with larger size, the optical band gap roughly decrease with an increase in the size of the crystallites (except that the samples annealed at 4500C does not follow this).

  Since the band gap of ZnSnN2 is still in argument, there is difficulty in correlating the experimental band gap quantitatively with the theoretical formula based on quantum size effect [J. Phys. Chem. 95(1991)525-532]. Therefore, we do not mention quantum size effect while discussing the optical band gap.

4     4) There is no Figure 5(a) to which authors refer in the text (page 9, top) discussing the experimental data on I-V curves of the Schottky diodes.

Reply:  The error is corrected.

Authors should check English grammar, e.g. subject-verb matching in a sentence in some cases.

Reply: The whole manuscript is checked carefully and the found errors are corrected.

Reviewer 2 Report

The focus of this work (nanocrystalline ZnSnN2 for device applications) is particularly of interest, although remains far from the usual inorganic materials commonly used. However, due to their interesting properties, Zn/N2 based materials with a IV atomic element, opens some opportunities to be explored as a possible materials for niche electronic applications. The present work, focusing in particular the ZnSnN2 (in nanocrystals) has some relevant features, particularly exhaustive in characterization and some device fabrication and analysis and, for sure, can be of interest. Nevertheless, some discussions – in particular regarding the devices behavior – follows some physical models and attempt to explain the experimental data in a way that, for me, lacks of some mode deep physical support. Additionally, absence of some data and some important experimental tests together with a non-exploration of more suitable physical models, leads to some incongruence in the final discussions, that, in my opinion, prevents the manuscript from being published without a thorough review. Particularly, the authors should pay attention to the following issues:

a) Please explain how the ITO resistivity was obtained. Also, includes the reference of the silver past used.

b) A more detailed explanation about Hall measurement electrical mobilities needs to be addressed (even in the supplementary information) in particular the controverse between n-/p-type nature of the materials.

c) Avoid sentences like “For the Schottky diodes, the forward-biased voltage is defined as the state at which the Ag electrode in direct contact with the ZnSnN2 layer has higher electrostatic potential than the other electrode which is in direct contact with ITO”. Just explain where is the anode and the cathode and (by the way) give one simple explanation why each interface appears to be rectifier in relation to the other (use the energy diagram for suggestion). All energy diagrams needs to have the respective energy values explicitly included.

d) In page 5, a more physical explanation needs to be done regarding the sentence “The fact that no reversal in the conduction type is observed in the Hall effect measurement of our samples very possibly results from the nanocrystalline grains which provide limited long-range order”. Include some references that could support such affirmation.

e) In page 8, the authors says that “…since the structure of Ag\ITO\ZnSnN2\ITO\Ag shows linear JV curves”. It is a little strange and no possibility of a double opposite diode was explored (please explain also in more detail how such structure was made). Moreover, plot the figure S5 in log scale for a better comparison with the figure 6.

f) All the discussion of the J-V data needs, under SCLC conditions, be done considering the usual models (Child Law, Mott-Gurney approach – with impacts on the electrical mobility – and the Mark-Helfrich domain). This implies experimental data (J) for applied voltages over 1V; otherwise, the explanations done by the authors lacks the essential of the electrical transport nature (in the bulk-limited region) leading to (eventually) erroneous explanations. Moreover, under SCLC, the authors can easily estimate the electrical mobility and compare to the obtained by Hall measurements, Particularly of high importance, is the electrical mobility in the trap-dependent region of the applied voltage (Mott-Gurney and Mark-Helfrich models that can estimated the effective electrical mobility) and in the pure Child Law regime at more high applied voltages when the J will depends on V^2 (again but with an electrical mobility independent on the trap energy states).

g) From the data plotted in the figure 9, it seems that the model obeys to a deep trap regime, with a presence of deep and shallow traps. Estimating the respective energy levels, a more precise and suitable model for the electrical behavior of all devices ca be obtained.

h) In the C-V data (particularly visible in the Mott-Schottky plot in figure 7) it seems to be (with an high degree of possibility) that the device have two energy levels distribution that accounts to the electrical charge accumulation (two slopes in the 1/C^2 vs. V for V<0) one corresponding to a one state and another to the sum of both. The authors, before trying to explain the data using some models that are suspicious for a non-pure crystal material, needs to explore such hypothesis. Also, this can be related with the issue pointed in the previous point.

i) The authors needs to explain why C-V data was obtained for a frequency of 10kH. However, much more important, is to made a simple impedance spectroscopy of the samples and, not only can justify such frequency but also can easily modulated the data into electrical equivalent circuits (parallel R and CPE). In this approach, a definitive model of the injection / transport of electrical charges, can be made.

j) A more detailed discussion of the photovoltaic devices figures of merit needs to be done. Particularly of importance, if the adjust to an equivalent electrical circuit, extraction the parallel resistance (defects related) and the estimation of the photogenerated electrical current Jph. The J0/Jph ratio and the defects influence in the transport / recombination (loss) and electrodes charge capture efficiency, can gives valuable information about the electrical properties of the devices made on ZnSnN2.

k) Finally, please check carefully all the manuscript, correcting some small English mistakes; also, some sentences should be rewritten in order to make them more understandable.

Author Response

Response to Reviewer 2

The focus of this work (nanocrystalline ZnSnN2 for device applications) is particularly of interest, although remains far from the usual inorganic materials commonly used. However, due to their interesting properties, Zn/N2 based materials with a IV atomic element, opens some opportunities to be explored as a possible materials for niche electronic applications. The present work, focusing in particular the ZnSnN2 (in nanocrystals) has some relevant features, particularly exhaustive in characterization and some device fabrication and analysis and, for sure, can be of interest. Nevertheless, some discussions – in particular regarding the devices behavior – follows some physical models and attempt to explain the experimental data in a way that, for me, lacks of some mode deep physical support. Additionally, absence of some data and some important experimental tests together with a non-exploration of more suitable physical models, leads to some incongruence in the final discussions, that, in my opinion, prevents the manuscript from being published without a thorough review. Particularly, the authors should pay attention to the following issues:

  1. a) Please explain how the ITO resistivity was obtained. Also, includes the reference of the silver past used.

Reply: The ITO resistivity was calculated by a multiplication of the sheet resistance (10.4Ωper square) and the thickness (597.1nm). The silver paste was from Uninwell and the product no. is 6778.

  1. b) A more detailed explanation about Hall measurement electrical mobilities needs to be addressed (even in the supplementary information) in particular the controverse between n-/p-type nature of the materials.

Reply:

Since the Raman spectra of ZnSnN2 is similar to its phonon density of states, there is difficulty in identifying the exact percentage of the amorphous and crystalline fractions. It is found that the mobility of the samples annealed at 300 0C is comparable with that of amorphous Si [Physica 117B&118B(1983)908-913]. After higher temperature annealing, the mobility increases up to 5.54 cm2V-1s-1. This is possibly because the percentage of crystalline fraction increases greatly and the transport property approaches that of crystalline semiconductors. In microcrystalline Si, crystalline semiconductor theory can give a good interpretation of the transport experiments if the crystalline volume fraction is high [Physica 117B&118B(1983)908-913].

   Theoretically, the n-type conduction and the relatively higher electron density of ZnSnN2 results from the fact that the intrinsic antisite donor defect of SnZn(Sn atoms occupy that of Zn positions) has the lowest formation energy among all the intrinsic defects and substitutional oxygen impurity [Adv. Mater. 26(2014)311,Adv. Mater. 31(2019)1807406]. The as-deposited ZnSnN2 samples are n-type conductive and remain to be n-type conductive after annealing. This agrees with the theoretical work [Adv. Mater. 26(2014)311,Adv. Mater. 31(2019)1807406]. In these ZnSnN2, we think that the transport properties are controlled by the crystalline fraction, namely, nanocrystalline grains providing limited long-range order and therefore these samples are naturally n-type conductive.

  1. c) Avoid sentences like “For the Schottky diodes, the forward-biased voltage is defined as the state at which the Ag electrode in direct contact with the ZnSnN2 layer has higher electrostatic potential than the other electrode which is in direct contact with ITO”. Just explain where is the anode and the cathode and (by the way) give one simple explanation why each interface appears to be rectifier in relation to the other (use the energy diagram for suggestion). All energy diagrams needs to have the respective energy values explicitly included.

Reply: Sentences like “For the Schottky diodes, the forward-biased voltage is defined as the state at which the Ag electrode in direct contact with the ZnSnN2 layer has higher electrostatic potential than the other electrode which is in direct contact with ITO” was revised as follows:

 "the forward-biased voltage is defined as the state at which the Ag electrode in direct contact with the ZnSnN2 is connected with the anode of the voltage".

  The structure of the Schottky diodes is as follows. It is show that the contact between Ag (labelled as 1 in the following figure.) and ITO and the contact between ITO and ZnSnN2 are Ohmic (Fig. S3, Supporting materials). Therefore, rectification comes from the contact between ZnSnN2 and Ag (labelled as 2 in the following figure). Actually, the Ag on ITO (labeled as 1 in the following) can be removed.

The structure of the solar cells is as follows. The contact between Ag and ITO and the contact between ITO and Cu2O are Ohmic. The Ag on ITO (labeled as 1 in the following) can also be removed.

Energy values are added to the energy diagrams in the manuscript and in the Supplementary Materials. 

  1. d) In page 5, a more physical explanation needs to be done regarding the sentence “The fact that no reversal in the conduction type is observed in the Hall effect measurement of our samples very possibly results from the nanocrystalline grains which provide limited long-range order”. Include some references that could support such affirmation.

Reply:

In Ref. 22 ([Physica 117B&118B(1983)908-913]), the authors use this to explain why no reversal in the conduction type is observed in their samples.

''[22]" is added to the end of that sentence.

  1. e) In page 8, the authors says that “…since the structure of Ag\ITO\ZnSnN2\ITO\Ag shows linear JV curves”. It is a little strange and no possibility of a double opposite diode was explored (please explain also in more detail how such structure was made). Moreover, plot the figure S5 in log scale for a better comparison with the figure 6.

Reply:

   The structure of Ag\ITO\ZnSnN2\ITO\Agis not a double opposite diode. Due to the fact that ITO is highly conductive, the contact between ITO and ZnSnN2 and the contact between ITO and Ag are Ohmic.

   Fig.S5 (now Fig.Sx) is replotted in log scale. Please refer to the Supplementary Materials.

  1. f) All the discussion of the J-V data needs, under SCLC conditions, be done considering the usual models (Child Law, Mott-Gurney approach – with impacts on the electrical mobility – and the Mark-Helfrich domain). This implies experimental data (J) for applied voltages over 1V; otherwise, the explanations done by the authors lacks the essential of the electrical transport nature (in the bulk-limited region) leading to (eventually) erroneous explanations. Moreover, under SCLC, the authors can easily estimate the electrical mobility and compare to the obtained by Hall measurements, Particularly of high importance, is the electrical mobility in the trap-dependent region of the applied voltage (Mott-Gurney and Mark-Helfrich models that can estimated the effective electrical mobility) and in the pure Child Law regime at more high applied voltages when the J will depends on V^2 (again but with an electrical mobility independent on the trap energy states).

Reply:

  The JV data in Fig.6 is analyzed. There is difficulty in correlating the SCLC models with the data quantitatively since the data does not follow the order of Ohmic conduction, , trap-filled limited region and . The following figures and discussion are inserted to the main text. Correspondingly, the figure caption of Fig.6 is revised.

  Fig.6(b) and (c) shows the logarithmic J-V curves of the samples without annealing or annealed at 300 0C. For both samples, both the forward and reverse biased data obey the power law ( ) but m varies in different voltage ranges. For the samples without annealing, below 0.18V, m is less than 1 and this suggests that the current is controlled by the diode. Between 0.18V and 0.68V, m is about 19.2. After 0.68V, m is about 230. This implies that after 0.18V, the current is bulk-controlled and there is exponential distribution of trap-levels within the forbidden gap of ZnSnN2. The reverse-biased data of the samples without annealing is diode-controlled since m is below 1 in the whole measurement range. For the samples annealed at 300 0C, m is well over 2 in both the forward and backward bias and this suggests that the current is bulk-controlled and annealing reduces the resistivity of ZnSnN2.  

  1. g) From the data plotted in the figure 9, it seems that the model obeys to a deep trap regime, with a presence of deep and shallow traps. Estimating the respective energy levels, a more precise and suitable model for the electrical behavior of all devices ca be obtained.

Reply: The following paragraph is inserted into the manuscript to give more discussion about Fig.9(a).

If we suppose that Vtr (0.19 V in Fig.9(a)) is the turn-on voltage at which the space charge limited conduction takes place in ZnSnN2 and VTFL (0.98 V in Fig.9(a)) is the voltage required to fill the traps of ZnSnN2, the trap density Nt and the trap energy level Et are calculated to be 2.14×1016cm-3and 0.1eV since Vtr equals  and VTFL equals ,where n is the free carrier density (n=ND=1015cm-3), d is the film thickness (236nm), and other symbols have the same meaning as previously defined.

  1. h) In the C-V data (particularly visible in the Mott-Schottky plot in figure 7) it seems to be (with an high degree of possibility) that the device have two energy levels distribution that accounts to the electrical charge accumulation (two slopes in the 1/C^2 vs. V for V<0) one corresponding to a one state and another to the sum of both. The authors, before trying to explain the data using some models that are suspicious for a non-pure crystal material, needs to explore such hypothesis. Also, this can be related with the issue pointed in the previous point.

Reply:

Fig.7 (a) and (b) are replotted below. Only one slope is found in Fig.7(a) and from it, the donor density is calculated to be 1.2×1031cm-3 .From the two slopes in Fig. 7(b), the donor density is calculated to be 9.4×1030cm-3 and 1.2×1031cm-3. The donor density is too large and the model [J. Electroanal. Chem.228(1987)135] does not work here.

  1. i) The authors needs to explain why C-V data was obtained for a frequency of 10kH. However, much more important, is to made a simple impedance spectroscopy of the samples and, not only can justify such frequency but also can easily modulated the data into electrical equivalent circuits (parallel R and CPE). In this approach, a definitive model of the injection / transport of electrical charges, can be made.

Reply: The original purpose of using 10 KHz is to measure also the effect of interface states since interface states might respond under relatively lower frequency. We are sorry that we do not have the impedance spectroscopy data.

  1. j) A more detailed discussion of the photovoltaic devices figures of merit needs to be done. Particularly of importance, if the adjust to an equivalent electrical circuit, extraction the parallel resistance (defects related) and the estimation of the photogenerated electrical current Jph. The J0/Jph ratio and the defects influence in the transport / recombination (loss) and electrodes charge capture efficiency, can gives valuable information about the electrical properties of the devices made on ZnSnN2.

Reply:The following paragraph is inserted to give a detailed discussion about the photovoltaic device.

To improve the conversion efficiency, the series resistance needs to be reduced and shunt resistance needs to be improved. The possible methods include better Ohmic contact and optimum film thickness, etc. The band gap needs to be reduced to improve the short-circuit current density Jsc. The theoretical open-circuit voltage Voc of the solar cell is estimated to be 0.24V since Voc approximately equal  if the series resistance and shunt resistance are negligible, and the photogenerated current equals the short-circuit current density Jsc. The diode current J0needs to be reduced to further improve the open-circuit voltage Voc.

  1. k) Finally, please check carefully all the manuscript, correcting some small English mistakes; also, some sentences should be rewritten in order to make them more understandable.

Reply:The whole manuscript is spell-checked and read carefully. The found errors were corrected.
